# Pan-Cancer Analysis of the TRP Family, Especially TRPV4 and TRPC4, and Its Expression Correlated with Prognosis, Tumor Microenvironment, and Treatment Sensitivity

**DOI:** 10.3390/biom13020282

**Published:** 2023-02-02

**Authors:** Zhenghao Chen, Youquan Zhao, Ye Tian, Rui Cao, Donghao Shang

**Affiliations:** Department of Urology, Beijing Friendship Hospital, Capital Medical University, Beijing 100050, China

**Keywords:** transient receptor potential channels, pan-cancer, TME, prognosis, immunotherapy

## Abstract

Background: Transient receptor potential (TRP) channels are involved in various physiological, pathological, and tumorigenesis-related processes. However, only a few studies have comprehensively analyzed TRP family members and their association with prognosis and tumor microenvironment (TME) in various cancers. Thus, in this study, we focused on TRP channels in pan-cancer and screened two typical TRP channels, TRPV4 and TRPC4, as examples. Methods: Based on the latest public databases, we evaluated the expression level and prognostic value of TRP family genes in pan-cancer tissues via various bioinformatic analytical methods, and investigated the relationship between the expression of TRP family genes with TME, stemness score, immune subtype, drug sensitivity, and immunotherapy outcome in pan-cancer tissues. Results: Pan-cancer analysis revealed that the TRP family genes were differentially expressed in tumor and para-carcinoma tissues. A significant correlation existed between the expression of TRP family genes and prognosis. The expression of TRP family genes was significantly correlated with stromal, immune, RNA stemness, and DNA stemness scores in pan-cancer tissues. Our results indicated that the expression of TRP family genes correlated with the sensitivity to various drugs including PLX-4720, SB-590885, and HYPOTHEMYCIN, immunotherapy outcome, and immune-activation-related genes. Immunohistochemical analysis revealed significant differential expression of TRPV4 in bladder and para-carcinoma tissues. Conclusions: Our study elucidated the possible role of TRP family genes in cancer progression and provided insights for further studies on TRP family genes as potential pan-cancer targets to develop diagnostic and therapeutic strategies.

## 1. Introduction

Transient receptor potential (TRP) channels are a series of ion channels responsible for various cellular functions. Originally discovered in drosophila, these channels were further observed to be widely distributed in mammals. Currently, the mammalian TRP-channel superfamily can be divided into six subfamilies: TRPC (canonical), TRPV (vanilloid), TRPM (melastatin), TRPP (polycystin), TRPML (mucolipin), and TRPA (ankyrin). Each of these subfamilies consists of many subtypes. TRP channels are a functional ion channel complex composed of four functional subunits. The complex can be a homo- or heterotetramer [1], but it is generally permeable to Ca^2+^. Its responses vary and can be activated by physical (depolarization, heat/cold, and mechanical stress) or chemical (pH and osmolarity) stimuli or by specific agonists.

The process of transformation, evolution, and tumor progression from normal cells to tumorigenic cells involves a complex network [2]. Previous studies have reported that TRP channels are involved in various physiological and pathological processes [3,4]. Yang D et al. mentioned in their review that the mutations in TRP channel genes result in abnormal regulation of TRP channel function or expression, and interfere with normal spatial and temporal patterns of intracellular local Ca^2+^ distribution. The resulting dysregulation of multiple downstream effectors, depending on Ca^2+^ homeostasis, is associated with hallmarks of cancer pathophysiology, including enhanced proliferation, survival, and invasion of cancer cells [5]. In the past 10 years, the role of TRP channels in tumorigenesis and the development of tumors has been gradually clarified. The existing evidence suggested that these channels induce tumor progression and invasion by participating in cell proliferation, abnormal differentiation, and autophagy processes [6,7,8]. Studies have reported that cancer-related changes in Ca^2+^ flux at the plasma membrane are diverse, and tumor cells can exploit this change to maintain tumor progression or resistance to therapy. Importantly, TRP-channel-mediated Ca^2+^ infiltration is associated with increased apoptosis resistance and metastatic potential, which results in activation of antiapoptotic and mitogenic pathways, establishment of antioxidant defense systems, initiation of autophagy, increased motility, and secretion of matrix metalloproteinases [9]. TRPV4 was reported to regulate calcium influx and release in HepG2 and hepatoblastoma cells [10]. In addition, Potter DA et al. reported that HGF/SF activated TRPV4 and TRPV1 channels and progressively amplified signaling, leading to cell motility and migratory phenotypes through reorganization of the actin cytoskeleton [11]. The changes in TRPV4 expression in bladder cancer tissue and para-carcinoma tissue were not very clear [12,13]; however, TRPV4 definitely detected mechanical and chemical stimuli, induced calcium influx, and promoted ATP release [14].

Therefore, further exploring the physiological functions of TRP channels and molecular mechanisms involved in cancer progression can enhance the understanding of the biological behavior of tumors and provide a basis for developing new therapeutic molecules against tumors.

In this study, we comprehensively analyzed the prognostic value of TRP family genes (TRPC, TRPV, TRPM, TRPP, TRPML, and TRPA) in pan-cancer tissues using the expression data downloaded from The Cancer Genome Atlas (TCGA). Further, we assessed the association between the expression of TRP family genes and tumor microenvironment (TME), immune subtypes, drug sensitivity, and immunotherapy outcome in patients with cancer. Moreover, we analyzed the correlation of two members of the TRP family, namely TRPV4 and TRPC4, with tumor mutational burden (TMB) and microsatellite instability (MSI), as well as the genes related with immune activation.

## 2. Materials and Methods

### 2.1. Identification of Differential Expression of TRP Family Genes in Human Pan-Cancer Tissues

RNAseq (FPKM) gene expression, clinical and pathological data, immune subtypes, survival data, and stemness scores (DNA methylation and RNA-based) for 33 cancers were downloaded from the online databases of UCSC Xena (http://xena.ucsc.edu, accessed on 19 June 2022 [15]. For pan-cancer TCGA analysis, the expression levels of 28 TRP family genes (Appendix A) were extracted and integrated using Perl (Appendix A), and the significant difference between tumor and para-carcinoma tissues was calculated using the Wilcox test. Boxplots and heatmaps were plotted using the R packages “ggpubr” and “pheatmap”, respectively. Correlation analysis of TRP family genes was performed using the R package “corrplot”. Additionally, it should be noted that the number of normal control samples of some cancers is less than five in the TCGA database. Too few samples will cause great systematic errors. Thus, in the process of writing the code, the data of these cancers were excluded when we compared the tumor and normal tissue.

### 2.2. Survival Analyses Based on the Expression Level of TRP Family Genes in Human Cancer

Survival data for each sample were extracted from the TCGA database to further analyze the relationship between the expression of TRP family genes and clinical outcomes. Overall survival (OS) was adequately assessed [16]. Survival analysis was performed using Kaplan–Meier (KM) survival curves and log-rank tests (the critical value of *p* was set to 0.05). The cutoff value was selected by the median expression level of TRP family genes in each cancer, thereby dividing each patient into a high- or low-risk group. Survival curves were plotted according to high- and low-risk groups using the “survminer” and “survival” R package methods. In addition, we performed Cox analysis to determine the relationship between the expression of TRP family genes and overall cancer prognosis. Finally, the R packages “survival” and “forestplot” were used to draw a forest plot.

### 2.3. Correlation Analysis of the Expression of TRP Family Genes with TME and Stemness Score in Pan-Cancer Tissues

We used the “estimate” and “limma” R packages to calculate the stromal and immune cell scores for predicting the infiltration of stromal and immune cells in pan-cancer tissues. Correlations between the expression of TRP family genes and RNA stemness scores (RNAss) as well as DNA stemness scores (DNAss) were analyzed using the Spearman method with the R packages cor. Test and limma. These two metrics were visualized using the R package “corrplot”.

### 2.4. Correlation Analysis of TRP Family Genes with Drug Sensitivity and Immune Subtypes

We downloaded the drug sensitivity processed data from the CellMiner data set (https://discover.nci.nih.gov/cellminer/, accessed on 19 June 2022). Data analysis and visualization of the results were processed using impute, limma, and ggplot2 in R software (version 4.0). The immune subtypes data were downloaded from UCSC. The correlation of TRP family and immune subtypes was mainly analyzed using limma and reshape2 in R software.

### 2.5. Correlation Analysis of TRP Family Genes and Immunotherapy Outcome

We downloaded the immunotherapy outcome data from the GES78220 and Imvigor210 datasets. The immunotherapy outcome of the samples is listed in Appendix A. Data analysis and visualization of the result were processed using limma, ggplot2, and ggpubr in R software

### 2.6. Correlation Analysis of TRPC4 and TRPV4 Expression with TMB and MSI

The TMB in tumor cells promotes immune recognition and correlates with the effectiveness of immunotherapy. An MSI occurs when new alleles are inserted into a tumor as a result of an alteration in microsatellites and is considered one of the hallmarks of immune-checkpoint-related therapy. These scores were computed from somatic mutation data obtained from TCGA. Two radar legends were generated to illustrate the relationship between TRPV4 and TRPC4 expression and TMB and MSI, respectively, based on Spearman’s rank correlation analysis. Additionally, coexpression analyses were performed on TRPC4 and TRPV4 with immune-activation-related genes.

### 2.7. Tissue Specimens and Immunohistochemistry

A total of 7 paired bladder cancer specimens and para-carcinoma tissue samples were obtained from the Beijing Friendship Hospital, Capital Medical University (Beijing, China), between January 2022 and March 2022. The clinical BLCA specimens were collected with permission from our Institutional Research Ethics Committee (NO.2021-P2-159). The immunohistochemical analysis was conducted as mentioned in our previous paper [17]. All samples were clinically and histologically diagnosed to be BLCA and were blindly stained by pathologists and evaluated in ImageJ software with a fixed set of operation. Expression levels of TRP genes were quantified as parameters after normalized ImageJ measurements and compared accordingly [18]. The antibody was acquired from abcam (ab307444) and a negative control was conducted during our tests.

### 2.8. Statistical Analyses

Student’s t test was used to assess the statistical significance between the two groups. Depending on the types of data, one-way ANOVA or Kruskal–Wallis tests were performed for the variables divided into more than three groups. The survival rates were calculated and visualized using KM survival curves. The significant differences were tested using the log-rank test. Pearson’s correlation analysis was performed to calculate the correlation coefficients. Univariate Cox proportional hazard models were used to determine the hazard ratios of variables and whether those variables were independent prognostic factors. *p* < 0.05 was considered significant.

## 3. Results

### 3.1. Expression and Correlation of TRP Family Genes in Pan-Cancer Tissues

We assessed the expression of TRP family genes in 33 types of cancers (Figure 1A). A series of genes in the TRP family including TRPC1, TRPC4, TRPC6, TRPM2, TRPM4, TRPM7, TRPV2, TRPV4, MCOLN1, MCOLN2, MCOLN3, PKD2, and PKDL1 were highly expressed in all cancer types. The correlation between various TRP genes was explored (Figure 1B). TRPV1 and PKD2, as well as TRPV6 and TRPM8, were the genes with the most significant positive correlation, whereas TRPC1 and TRPM4 were the most negatively correlated genes. We further explored the expression of all TRP family genes in 33 types of cancers (Figure 1C). TRPM4 was highly expressed in CHOL, and TRPM6 exhibited significantly lower expression in pan-cancer tissues, particularly in COAD and READ (Figure 1C).

### 3.2. Different Expression and Correlation of TRP Family Genes and Prognosis in Pan-Cancer Tissues

Further, we obtained the expression of TRP family genes from the TCGA database; the expression matrix is shown in Appendix A, and the differently expressed TRP family genes across all tumor and para-carcinoma tissues are shown in Figure 2A,E and Appendix A. Most tumors exhibited differential expression of TRP family genes in tumor and para-carcinoma tissues.

We subsequently screened 33 types of cancer and selected those including more than five para-carcinoma and tumor tissues to analyze the correlation between the expression levels of each TRP family gene and the prognosis of patients. Finally, 28 cancers were included in the analysis, and the KM survival curves of TRPC4 and TRPV4 are shown as an example in Figure 2B–D,F,G, with the rest of the survival curves of all the TRP channels in pan-cancers shown in Appendix A. Furthermore, we investigated the prognostic risk of TRP family genes, taking TRPC1, C4, C5, C7, V4, M1, and MCOLN1 as examples, in pan-cancer tissues using Cox regression analysis (Figure 3), and the Cox results of other TRP family members are listed in Appendix A.

### 3.3. Association of TRP Family Genes with TME and Stemness Score in Pan-Cancer Tissues

TME played a key role in stimulating cancer cell heterogeneity, increasing multidrug resistance, and contributing to cancer progression and metastasis. Our previous study had identified a predictive role for the TRP family genes in pan-cancer tissues. It is very important to explore the relationship between the expression of TRP family genes and TME in pan-cancer tissues. The ESTIMATE algorithm was used to calculate the immune and stromal scores of pan-cancer tissues (Figure 4). The scores were significantly positively correlated with the expression levels of most TRP family genes (Figure 4A,B). Similarly, a significant positive or negative correlation existed between the expression of HER family genes and RNAss (Figure 4C) and DNAss (Figure 4D) in pan-cancer tissues. The correlation and *p* value of all the TRP family genes in pan-cancer tissues are listed in Appendix A.

Further, we explored the association between the expression of TRP family genes and immune, stromal, estimate, and stemness scores in selected types of cancer (BRCA and COAD) (Figure 5 and Figure 6). TRP family genes exhibited extensive correlation with TME as well as DNAss and RNAss in BRCA and COAD.

### 3.4. Association of TRP Family Genes with Immune Subtypes in Pan-Cancer Tissues

In a previous study, Thorsson et al. identified six immune subtypes (C1–C6) based on the immunogenomic analysis of more than 1000 tumor samples from 33 cancer types [19]. These categories were significantly associated with prognosis and genetic and immunomodulatory alterations in tumors. Thus, we further explored the correlation between TRP family genes and immune subtypes. TRPC4, TRPC6, TRPV2, and TRPV4 were differently expressed in BRCA and COAD (Figure 7). TRPC4, TRPC6, and TRPV2 exhibited higher expression in C6 in COAD and BRCA. TRPV4 was highly expressed in C6 and clearly less expressed in C4 in COAD. However, the expression of TRPV4 in BRCA was not significant.

### 3.5. Association of TRP Family Genes with Pan-Cancer Treatments

To explore the correlation between TRP family genes and drug sensitivity in various human cancer cell lines, drug sensitivity data were obtained from CellMiner. CellMiner is a database including 60 different cell lines and the sensitivity to more than 20,000 different drugs for each cell line. We mainly extracted and examined the relationship between all TRP channels and FDA−approved drugs or drugs in clinical trials. This resulted in a total of 860 kinds of drugs including, for example, Chelerythrine, Veliparib, Amuvatinib, Cytarabine, and so on. All the data downloaded from CellMiner, including the expression matrix of these cell lines, as well as the drug sensitivity, are listed in Appendix A. We listed 16 TRP family genes in Figure 8 and Appendix A, which were correlated with a certain drug. TRPM1 was positively correlated with the sensitivity to PLX-4720, SB-590885, and hypothemycin (Figure 8A,F,J), and negatively correlated with the sensitivity to varbulin (Figure 8G). TRPV2 was positively correlated with the sensitivity to PLX-4720, SB-590885, hypothemycin, and tipifarnib (Figure 8B,E,I,N). TRPC4 was negatively correlated with the sensitivity to sepantronium bromide, alvespimycin, and ONX-0914 (Figure 8C,D,O). Furthermore, TRPV4 exhibited positive correlation with the sensitivity to PLX-4720 and SB-590885. A negative correlation existed between the sensitivity to lexibulin and TRPV4 (Figure 8L,M,P).

In recent years, a dramatic shift has been observed from combined chemotherapy and radiotherapy to more precise immunotherapy. To further explore the correlation between TRP channels and immunotherapy response, we obtained the expression levels and immune response from the GSE78220 (Figure 9A and Figure 10A) and IMvigor210 (Figure 9B and Figure 10B) datasets. Considering the results of the differential expression of TRP family genes between tumor and para-carcinoma tissues and RNAss, DNAss, and TME scores, we selected TRPC4 and TRPV4 as the representative TRP family genes. Importantly, TRPV4 and TRPC4 were both negatively correlated with immunotherapy response.

TMB, which can be easily accessed to replace overall neoantigen detection, has been identified as a potential biomarker to predict the clinical outcome of immunotherapy [20,21,22]. In addition, MSI was reported to be correlated with immunotherapy outcomes [20,21,23]. Thus, we downloaded the TMB and MSI data from the TCGA database and explored the relationship between TMB/MSI and the expression of TRPV4 and TRPC4 (Figure 9C,D and Figure 10C,D, Appendix A). A significant correlation existed between the expression of TRPV4 and various cancers including BRCA, ESCA, HNSC, KIRC, LGG, LIHC, LUAD, PCPG, PRAD, SKCM, STAD, and THYM (Figure 9C). Meanwhile, a significant correlation was observed between TRPV4 expression and MSI in various cancers, including BRCA, COAD, LAML, LUSC, MESO, PAAD, PRAD, STAD, TGCT, and UVM (Figure 9D). The positive correlation between TRPC4 and TMB/MSI is shown in Figure 10C,D.

Furthermore, we explored the coexpression of immune-activation-related genes and TRPV4 (Figure 9E) and TRPC4 (Figure 10E). A significant correlation was observed between TRPV4/TRPC4 and immune-activation-related genes in almost all 33 types of cancer. In addition, we explored the expression of TRPV4 in our own patients’ tissue, and the result was similar to the conclusions above. In comparison to para-carcinoma tissues (Figure 11A,C), TRPV4 was highly expressed in bladder cancer tissues (Figure 11B,D,E).

## 4. Discussion

Cancer is one of the main causes of rising morbidity and mortality worldwide and has become a major burden and medical challenge in recent decades. Data obtained from experimental models and preclinical and clinical trials demonstrated the correlation between classical clinicopathological tumor markers and partial TRP channel expression in pan-cancer tissues. This suggested that TRP channels are valuable diagnostic and prognostic markers. At the same time, the inhibition or enhancement of various TRP channels have been reported to exhibit good antitumor effects in vitro and in vivo [24,25]. Some targeted drugs have a close relationship with TRP family genes [26,27,28]. Thus, TRP channels are expected to be studied for developing pan−cancer-targeted therapy. With the in-depth understanding of the role of TRP channels in pan−cancer tissues, they have been reported to be the therapeutic targets in certain tumors [27] and are correlated with tumor progression and metastasis [26,29,30]. For example, TRPV4 is one of the members of the TRPV channel, which can detect mechanical pressure, osmotic pressure (hypotonic), medium temperature (>27 °C), and chemical stimulation and is a nonselective cation channel. Its expression is detected in the esophagus, kidneys, liver, lungs, and bladder [31,32,33,34,35]. Li M et al. reported that the stimulation of native TRPV4 or transiently transfected TRPV4 in A375 cells (human melanoma cell line) could induce significant extracellular secretion, which acquired TRPV4-mediated calcium influx, as well as a series of key regulators of exocytosis including lysosome-associated proteins and multiple folding and vesicular transporters. By identifying a series of intracellular events after TRPV4 activation, Li M et al. demonstrated a critical role of TRPV4 in extracellular processes and calcium-mediated ferroptosis [36]. Additionally, administration of 4α-PDD (agonist of TRPV4) downregulated adhesion−related tumor suppressor genes in 4T07 (mouse breast cancer cell line). This indicated that TRPV4 increased the metastatic potential of tumors [37]. In bladder cancer, the inhibition of overexpressed TRPV4 significantly reduced E-cadherin expression. TRPV4-induced activation of AKT and FAK further affected E-cadherin expression [38].

In this study, we obtained the expression level of TRP family genes from the TCGA database and reported the differentially expressed TRP family genes in 33 types of cancer. We further explored the correlation of each TRP channel with prognosis using KM survival curves and Cox regression analysis and reported the prognostic value of TRP family genes. Moreover, we explored the correlation of the TRP family with TME and stemness scores in each cancer. We evaluated the distribution of the TRP family in C1–C6 pan-cancer immune subtypes and observed that TRPC4, TRPC6, TRPM4, TRPV2, TRPV4, MCOLN1, and PKD2L1 had the potential to predict the immune subtype. Thus, we further analyzed the correlation of TRPV4 and TRPC4 with drug sensitivity, immunotherapy response, TMB, MSI, and immune-activation-related genes. Therefore, our study provided insights into the use of TRP family genes as prognostic markers in cancer and contributed to the potential development of therapeutics involving TRP family genes.

In recent years, several studies were performed on single TRP channels; however, the role of the TRP family in pan-cancer tissues was still relatively unexplored. In our study, we comprehensively analyzed the expression levels of TRP family genes in various tumors (Figure 1). The expressions of TRPC1, TRPC4, TRPC6, TRPM2, TRPM4, TRPM7, TRPV2, TRPV4, MCOLN1, MCOLN2, MCOLN3, PKD2, and PKDL1 were generally high in pan-cancer tissues. TRPM4 was widely highly expressed in almost all the cancers (Figure 1C). This is consistent with previous studies on various tumors including colorectal cancer, breast cancer, and prostate cancer [39,40,41]. Moreover, it was shown to be a potential prognostic marker for cancer and a promising anticancer drug target candidate. Since TRPM4 is a Ca^2+^-activated monovalent cation channel, its ion conductivity can decrease intracellular Ca^2+^ signaling, leading to further interaction with different partner proteins. Thus, TRPM4 enabled many interventions in signaling pathways, increasing the possibility of drug development targeting TRPM4 [42,43]. TRPM2 exhibited similar results; higher expression of TRPM2 had been proven in various cancers including breast cancer, prostate cancer, pancreatic cancer, leukemia, and neuroblastoma [44]. Lin R et al. added PKC/MEK inhibitor to BxPC-3 cells overexpressing TRPM2 and demonstrated that TRPM2 might directly activate PKCα via calcium or indirectly activate PKCε and PKCδ by increasing DAG in pancreatic cancer, which promoted pancreatic cancer by the activation of the downstream MAPK/MEK pathway [45]. Ji D et al. analyzed TRPM2 as an ion channel in terms of oxidative stress and reported that it was essential for cellular function and played an important role in oxidative stress and inflammation; they summarized the current understanding of TRPM2 in brain tumors and reviewed potential pharmacotherapeutic roles of TRPM2. The importance of TRPM2, as a potential therapeutic target for brain tumors, in ion channels and pharmacology was reported in a previous study [46]. In our study, the differential expression of TRPM2 and TRPM4 in tumor and para-carcinoma tissues was extremely clear (Appendix A), indicating the important role of these TRP family genes in the tumorigenicity in various cancers. Moreover, KM survival curves indicated that the expression of TRP family genes was closely correlated with the prognosis of patients.

In recent years, the focus of cancer research has gradually shifted from tumor cell metastasis to surrounding core cancer cells, termed TME [47,48,49]. Immune and stromal cells are the two major nontumor components of TME and have been recognized as important in the diagnosis and prognostic assessment of tumors. The cells in TME and degree of infiltration of immune and stromal cells in the tumor have a significant impact on prognosis. The calculation of immune and stromal scores based on the ESTIMATE algorithm helps to quantify immune and stromal components in tumors. In this algorithm, immune and stromal scores are calculated by analyzing specific gene expression signatures of immune and stromal cells to predict nontumor cell infiltration. To better understand the prognostic correlation of immune and stromal cells with the TRP family, we systematically analyzed the expression of TRP family genes with the immune and stromal scores calculated. The expression of TRPV2 was clearly positively correlated with immune and stromal scores in most cancers (Figure 4), except KIRP, PCPG, THYM, and UCS, in which the correlation between stromal score and TRPV2 was not significant.

We assessed tumor stemness scores of TRP family genes in various cancers using DNAss and RNAss. The results indicated the negative role of the TRP family in characterizing cancer stem cells. To further explore the association between the TRP family and TME, stemness scores in certain types of cancer (BRCA and COAD) was analyzed. We conducted correlation analysis. The expression of TRP family genes (including TRPC1, TRPC4, TRPC6, TRPV2, TRPV4, MCOLN1, and PKD2L2) exhibited a significant positive correlation with stromal, immune, and ESTIMATE scores and negative correlation with DNAss and RNAss in BRCA and COAD (Figure 5 and Figure 6).

The CellMiner database is based on 60 cancer cell-types listed by the National Cancer Institute’s Center for Cancer Research (NCI). The NCI-60 cell line is currently the most widely used cancer cell sample population for anticancer drug testing. CellMiner is a free online tool that provides a centralized source of molecular and pharmacological characterization data for the widely studied NCI-60 cancer cell population [50]. In our study, we explored the correlation between the expression of TRP family genes and drug sensitivity through the data downloaded from CellMiner. Several drugs that were correlated with TRP family genes were obtained. Further, we observed that TRPC4 and TRPV4 correlated with immunotherapy response according to the data of the GSE78200 and IMvigor210 datasets, indicating the potential role of the TRP family in predicting immunotherapy outcomes. Furthermore, we confirmed the role of TRPV4 and TRPC4 in predicting immunotherapy prognosis by examining the correlation between the TRP family and immune-activation-related genes.

## 5. Conclusions

In summary, our study elucidated the expression profile of TRP family genes that correlated with disease prognosis, TME, stemness score, and the treatment outcome of various cancers. In addition, the expression level of TRP family genes in tumor cells correlated with the sensitivity to various drugs and the outcome of immunotherapy. These results can provide reference for further study on TRP family genes as potential pan-cancer targets to develop diagnostic and therapeutic strategies.

## Figures and Tables

**Figure 1 biomolecules-13-00282-f001:**
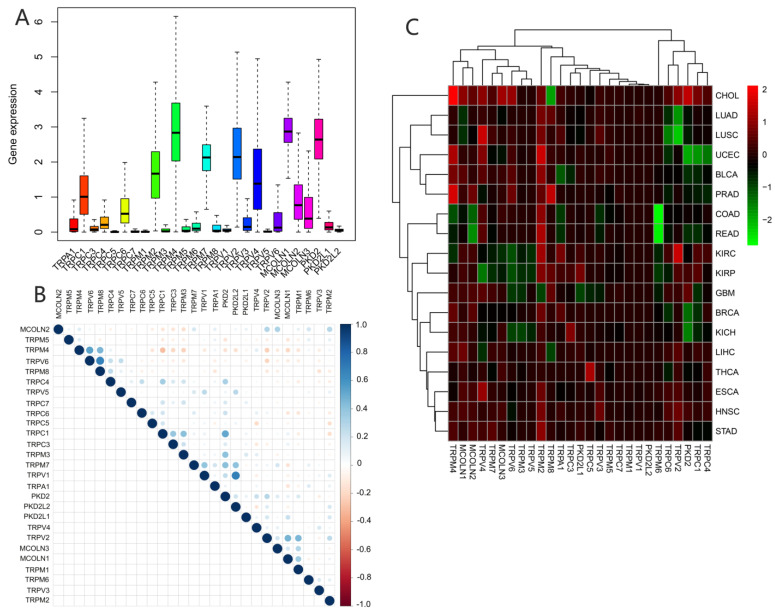
Expression levels and correlations between TRP family genes in various cancers from TCGA database. (**A**) Over- or underexpression of TRP family genes in various cancers. (**B**) Correlations between TRP family genes. Blue and red dots represent positive and negative correlations, respectively. (**C**) Expression data from TCGA database showing the expression of TRP family genes in various types of cancer. The color of each small rectangle represents high or low expression of TRP family genes in each cancer. Red and green indicate high and low expression, respectively.

**Figure 2 biomolecules-13-00282-f002:**
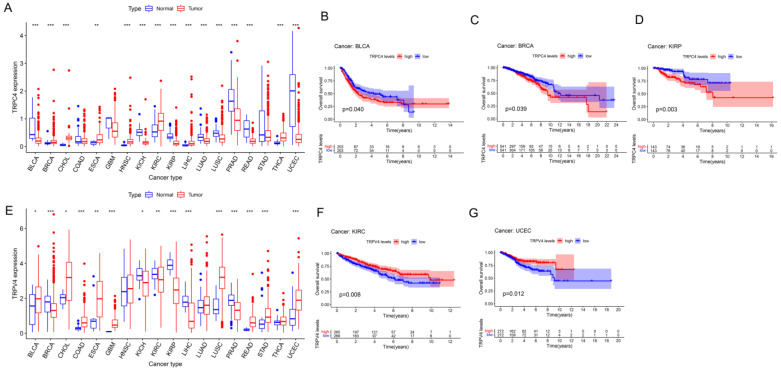
Different expression and Kaplan–Meier survival curves of TRPC4 and TRPV4 in pan-cancer (**A**). Differential expression of TRPC4, Kaplan–Meier survival curves of TRPC4 in (**B**) BLCA, (**C**) BRCA, and (**D**) KIRP. (**E**) Differential expression of TRPV4, Kaplan–Meier survival curves of TRPV4 in (**F**) KIRC and (**G**) UCEC. The red and blue rectangular boxes represent gene expression levels in tumor and normal tissues, respectively. * *p* < 0.05, ** *p* < 0.01, and *** *p* < 0.001. Red- and blue-colored names indicate high and low expressions of the corresponding TRP family genes, respectively.

**Figure 3 biomolecules-13-00282-f003:**
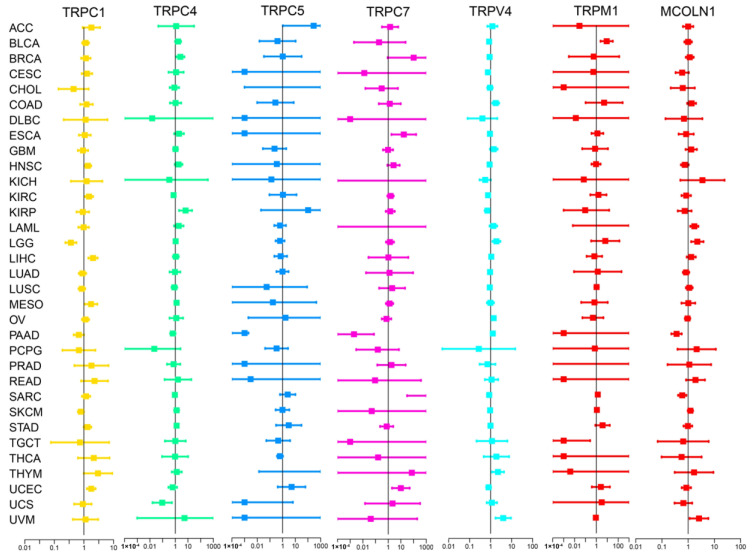
Cox regression analysis of the correlation between the expression of TRP family genes and survival. Lines with different colors represent the risk values for different genes within the tumor, with a hazard ratio <1 and >1 indicating low and high risk, respectively.

**Figure 4 biomolecules-13-00282-f004:**
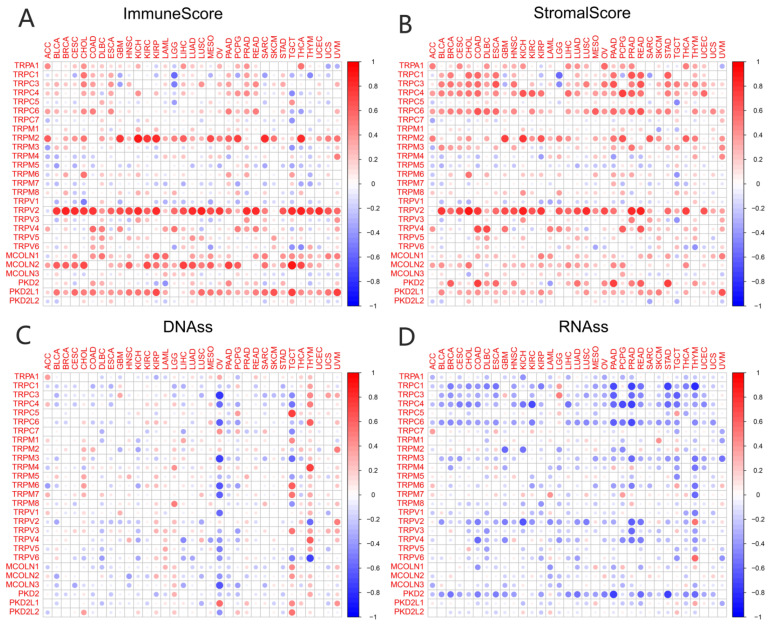
Correlation of the expression of TRP family genes with TME and stemness score in pan-cancer tissues. (**A**,**B**) I expression of TRP family genes correlated with various mesenchymal and immune cancer scores. Red and green dots indicate a positive and negative correlation between the gene expression and mesenchymal score, respectively. (**C**,**D**) The correlation between the expression of TRP family genes and RNAss and DNAss in pan-cancer tissues. Red and blue dots indicate a positive and negative correlation between the gene expression and immune score, respectively.

**Figure 5 biomolecules-13-00282-f005:**
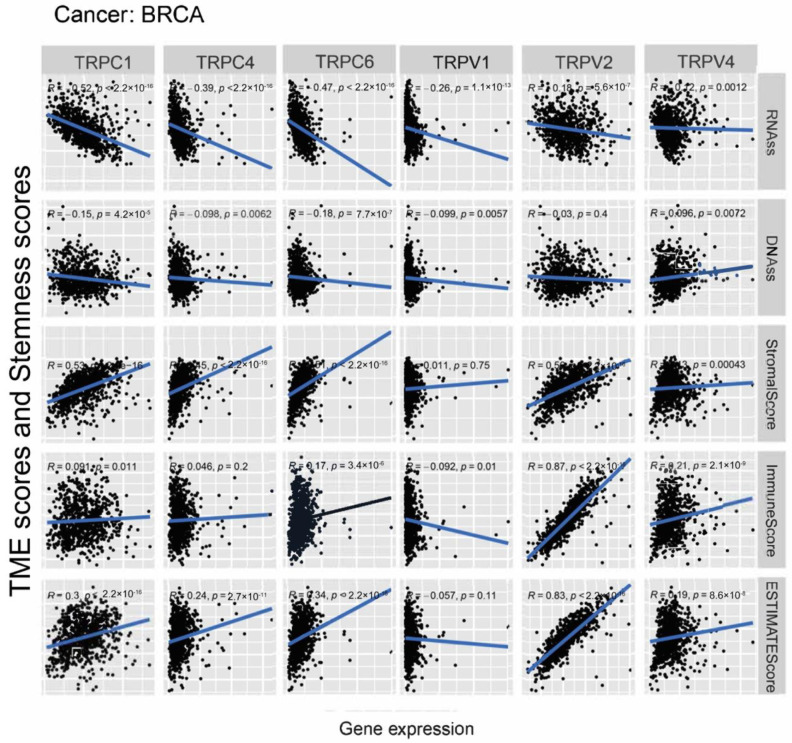
Correlation analysis of the expression of TRP family genes with RNAss, DNAss, stromal score, immune score, and ESTIMATE score in BRCA.

**Figure 6 biomolecules-13-00282-f006:**
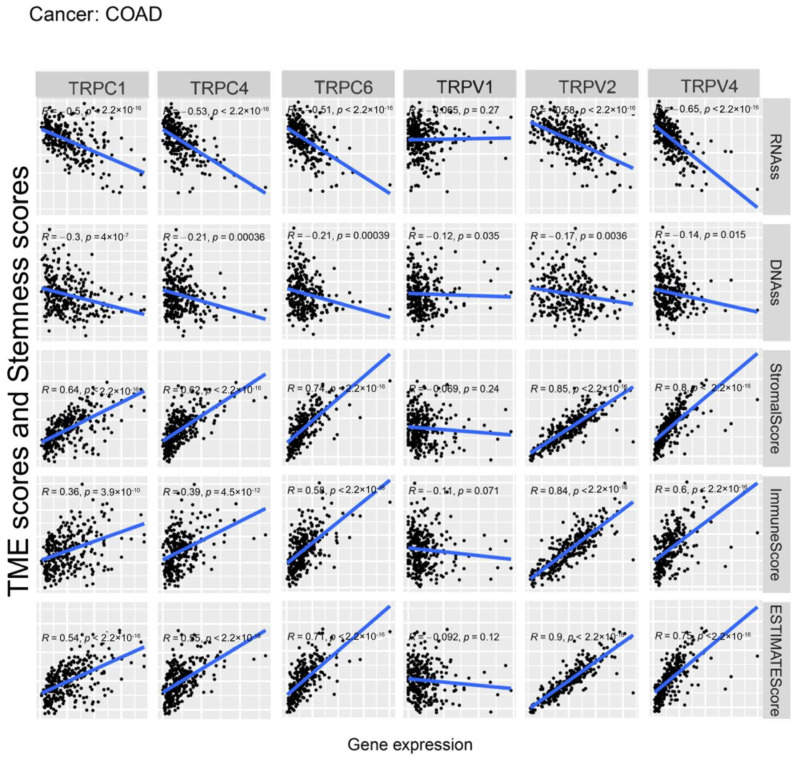
Correlation analysis of the expression of TRP family genes with RNAss, DNAss, stromal score, immune score, and ESTIMATE score in COAD.

**Figure 7 biomolecules-13-00282-f007:**
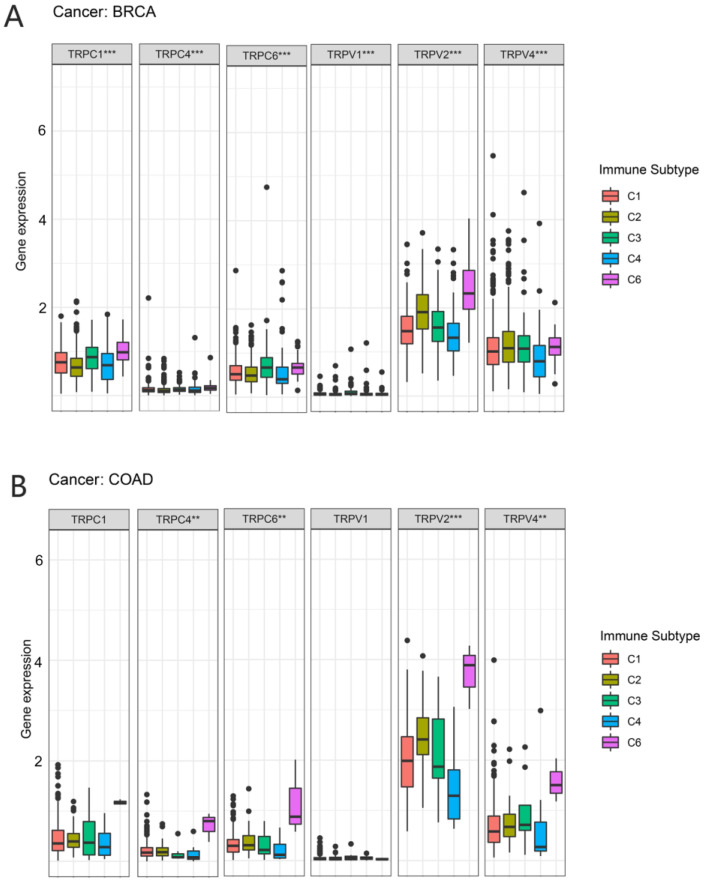
Correlation between the expression of TRP family genes and immune subtypes in BRCA and COAD. The significantly different expression of TRP family genes in various immune subtypes in (**A**) BRCA and (**B**) COAD. (**A**) TRPC1, TRPC4TRPC6, TRPV1, TRPV2, and TRPV4 were all differently expressed among the immune subtypes. (**B**) TRPC4TRPC6, TRPV2, and TRPV4 were differently expressed among the immune subtypes, especially the C6 subtype. X-axis represents immune subtype, and y-axis represents gene expression. C1, wound healing; C2, IFN-g dominant; C3, inflammatory; C4, lymphocyte depleted; C5, immunologically quiet; C6, TGF-β dominant. ** *p* < 0.01, and *** *p* < 0.001.

**Figure 8 biomolecules-13-00282-f008:**
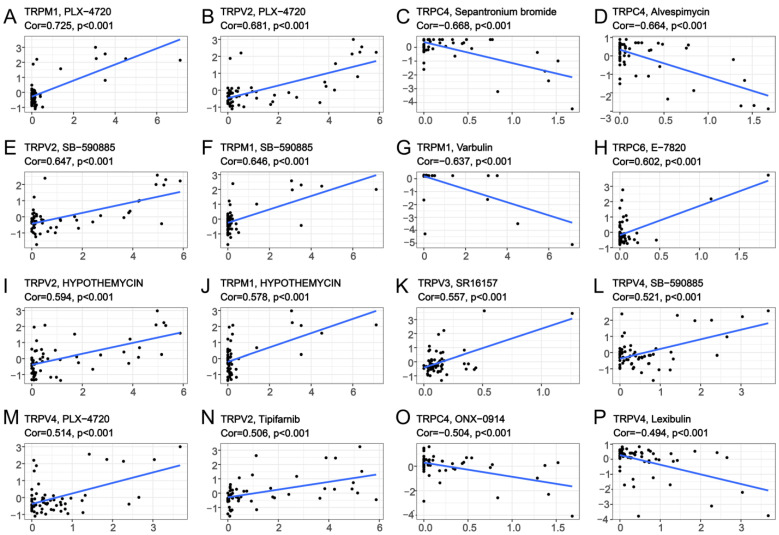
Drug sensitivity analysis of TRP family genes. The x-axes are the sensitivity of certain drugs and the y-axes are the expression level of certain TRP family members.

**Figure 9 biomolecules-13-00282-f009:**
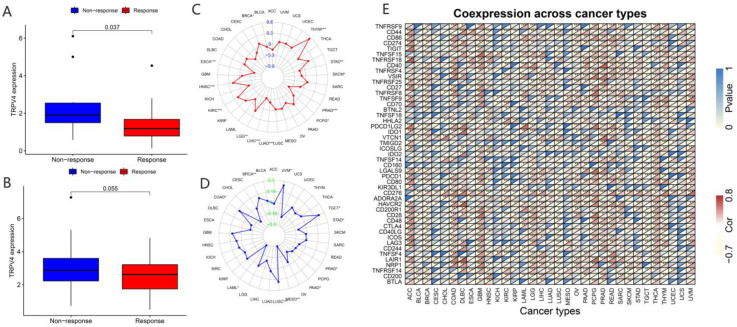
Correlation of immunotherapy outcome, TMB, MSI, and immune-activation-related genes with the expressions of TRPV4. * *p* < 0.05; ** *p* < 0.01; *** *p* < 0.001.

**Figure 10 biomolecules-13-00282-f010:**
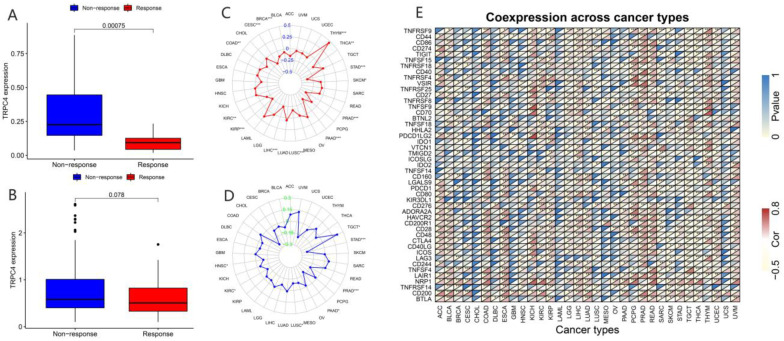
Correlation of immunotherapy outcome, TMB, MSI, and immune-activation-related genes with the expressions of TRPC4. * *p* < 0.05; ** *p* < 0.01; *** *p* < 0.001.

**Figure 11 biomolecules-13-00282-f011:**
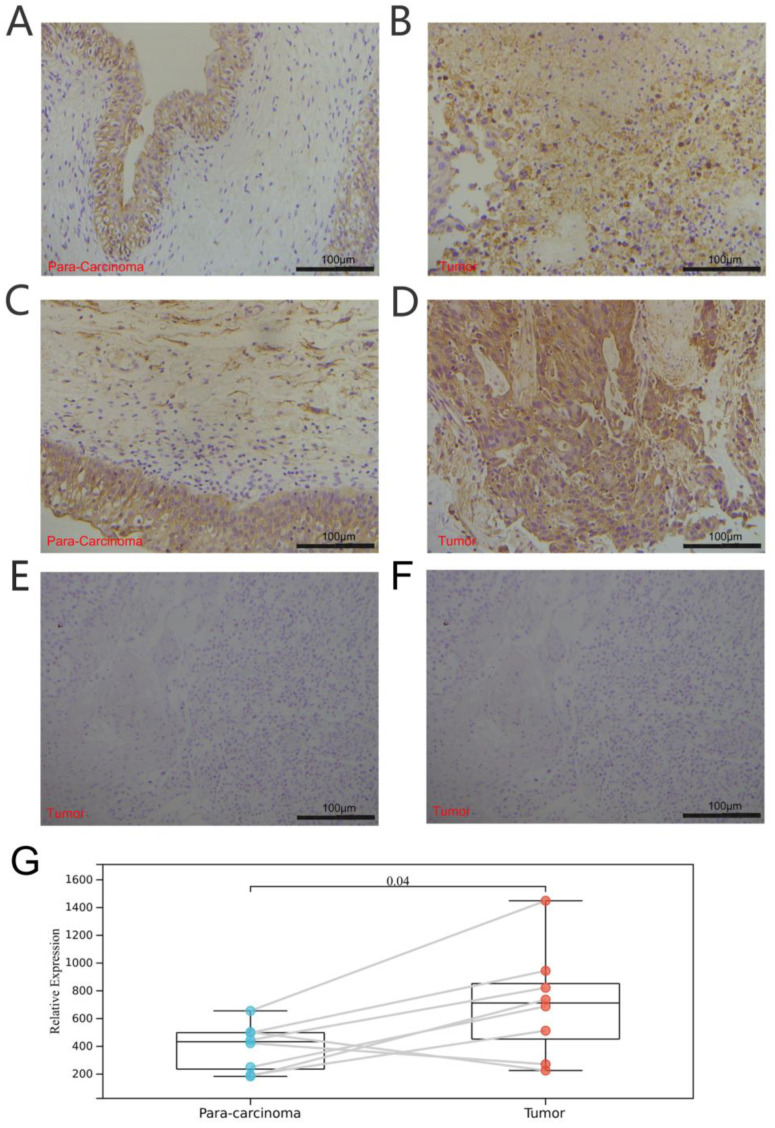
Immunohistochemical analysis of bladder cancer and para-carcinoma tissues. (**A**,**C**) Expression of TRPV4 in para-carcinoma tissues. (**B**,**D**) Expression of TRPV4 in bladder cancer tissues. (**E**,**F**) Negative control of para-carcinoma and bladder cancer tissues. (**G**) Paired comparison of TRPV4 in bladder cancer and para-carcinoma tissues.

## Data Availability

The datasets supporting the conclusions of this article are included in the article and Appendix A.

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
