# Peer review of "Pan-Cancer Analysis of the TRP Family, Especially TRPV4 and TRPC4, and Its Expression Correlated with Prognosis, Tumor Microenvironment, and Treatment Sensitivity"

_biomolecules, 2023, doi:10.3390/biom13020282_

Round 1
Reviewer 1 Report
Authors investigated the expression of TRP channels in the TCGA data set, however the storyline is quite unclear and must worked out better. Massive data is presented however its unclear how these data help drawing a conclusion, whether expression of TRPs is correlated with good or bad prognosis and why. Is expression of TRPs in all immune cell subtypes correlated with cancer progression and correlated with drug response? I would hardly recommend to select only some important TRPs that might be associated with a certain phenotype from the initial “screen” and see whether these are correlated with drug response. Regarding drug response: what kind of drugs are shown? Known inhibitors or TRP channels? The manuscript is not suitable for publication and authors should really consider whether putting a lot of data in a manuscript helps understanding the story.
Lane 42-46: How does this information refer to TRP? Are TRPs frequently mutated in cancers (as authors performed investigations based on TCGA data they might have checked).
Figures 1A-B: the labeling is hard to recognize, font size must be increased. In Fig. 1B I would suggest not showing the correlation scores, this makes the figure extremely unübersichtlich. Instead authors should provide a supplementary table showing the scores and associated p-values.
Line 145: The mentioning of R software at this point is useless as this belongs to standard procedures and must be mentioned in the Methods section.
Figure 2: a very busy figure showing a lot of information which makes it difficult to extract for the readership. Authors should focus on the most important results here showing and should provide the information of not expressed TRPs to the supplement. Moreover, authors should draw a conclusion and should focus on TRPs that are indeed higher expressed in cancer types as this should refer to more aggressive phenotypes.
Figure 3: Authors may combine figs 2 and 3, showing the levels of TRPs expression and the consequence. Other than stated/mentioned in the introduction, TRPs expression shows a context dependent function and is not necessarily associated with poor survival.
Figure 4: Like figure 2, extremely busy and scales of forest plots are nearly invisible. I hardly recommend that authors focus here on expressed TRPs, some plots should be moved to the supplement. Moreover, I am not sure why authors are showing these additional data as survival curves already demonstrate the relationship of TRPs expression and consequence for patients survival.
Figure 5: I am wondering whether TRPC2 expression is related to immune cells due to the high correlation with the immunscore in all cancers investigated. Authors might focus on that.
Figure 6 and 7: Should be simplified and focused on the most important TRPs such as TRP2 showing a significant correlation. Providing all this information should be avoided.
Figure 8: What is the conclusion here and what kind of immune subtypes are presented. This must be stated at least in the legend.
Figure 9: No axis labeling is provided and information about the drug targets is not provided. Authors should show the only the important information here and I really doubt the p-values given for the sometimes extremely low correlation.
Figure 10: I am sorry but this figure is useless as it´s nearly impossible to see any labeling and as the figure legend provides nearly no information/explanation what kind of plots are shown here.
Figure 11: What is stained here? TRPV4? And why switched authors to bladder cancer? What is the conclusion here? Is this a confirmed tumor? In E: how did authors derive these data? Expression data or quantification of IHC?
Author Response
Dear reviewer,
Thank you for reviewing our manuscript and for the constructive comments, which greatly helped us to improve the manuscript. We have heavily revised our manuscript.
Please see our answers in the attachment and many thanks to you for your precious comments.
Best Wishes!
Yours Chen

Reviewer 2 Report
Comments and Suggestions for Authors
Chen et al address the relevance of TRP channel in the context of TME and tumor prognosis of bladder cancer. Methodologies are based on the latest public databases, combined with various bioinformatics analysis. There is a limited data on levels of TRP channels in bladder cancer tissues.
However, the topic has relevance to better understand the TME of bladder cancer that will help in future diagnostics and therapeutics.
However, following comments will help to enrich the quality of paper.
Please check typo and grammatical errors.
In the abstract, objective and background of the paper do not synchronize with the method sections.
In method section, additional cells/tissue based assays such as western blot will strengthen the claim regarding TRP channels and bladder cancer.
Please check and make changes for abbreviations such as tumor microenvironment (TME). After first use of tumor microenvironment (TME), please use TME afterwards.
The authors need to justify reasons behind the use of IHC in bladder cancer and levels of TRP. However, the title of paper is on generalized “tumor microenvironment and treatment outcome”
Therefore, conclusion section does not fully coherence with title and method sections.
Author Response
Dear reviewer,
Thank you for reviewing our manuscript and for the constructive comments, which greatly helped us to improve the manuscript. We have heavily revised our manuscript.
Please see our answers in the attachment and many thanks to you for your kind comments.
Best Wishes!
Yours Chen

Round 2
Reviewer 1 Report
Dear authors,
thanks for considering my suggestions, the manuscript is not much more streamlined than before. However, a general advice: please keep in mind that the qualifity of your figures such as resolution, proper axes labeling and font size is very important. Hence, I am encouraging authors for checking the resolution of figures, I observed that the correlation map in figure 1, particularly the labeling is blurry.
Moreover, please A.) provide information about the immune subtypes shown directly in the figure legend either by axes labeling OR as a legend not both. A labeling C1, C2 etc. is not sufficient and does not provide any relevant information here. B.) Please improve IHC images shown in figure 11. They are quite dark and its likely they are becoming more dark in the final proof. As imges does not provide any information such as the tissue type, where normal and tumor tissue is shown its hard to judge anything here. Scale bars are also missing. So can you please indicate the type of tissue (tumor or stroma) directly in the images? Has the tumor character been confirmed by a pathologiest and have the antibody been tested/established on control sections?
Author Response
Dear reviewer,
We really thank you again for taking the time to review this paper a second time, and your comments are really helpful and enlightening to us. And our answer to the comments were listed in the attachment.
Best wishes!
Chen
